# Improvement in Infection Prevention and Control Performance Following Operational Research in Sierra Leone: A Before (2021) and After (2023) Study

**DOI:** 10.3390/tropicalmed8070376

**Published:** 2023-07-23

**Authors:** Senesie Margao, Bobson Derrick Fofanah, Pruthu Thekkur, Christiana Kallon, Ramatu Elizabeth Ngauja, Ibrahim Franklyn Kamara, Rugiatu Zainab Kamara, Sia Morenike Tengbe, Matilda Moiwo, Robert Musoke, Mary Fullah, Joseph Sam Kanu, Sulaiman Lakoh, Satta Sylvia T. K. Kpagoi, Kadijatu Nabie Kamara, Fawzi Thomas, Margaret Titty Mannah, Victoria Katawera, Rony Zachariah

**Affiliations:** 1National Infection Prevention and Control Coordinating Unit, Ministry of Health and Sanitation, Freetown 00232, Sierra Leone; christy.conteh@yahoo.com (C.K.); ramatungauja@yahoo.com (R.E.N.); 2World Health Organization Country Office, Freetown 00232, Sierra Leone; ibrahimfkamara@outlook.com (I.F.K.); musoker@who.int (R.M.); katawerav@who.int (V.K.); 3Centre for Operational Research, International Union Against Tuberculosis and Lung Disease, 75001 Paris, France; pruthu.tk@theunion.org; 4US Centre for Disease Control and Prevention Country Office, Freetown 00232, Sierra Leone; rugiatuzkamara@gmail.com; 5Ministry of Health and Sanitation, Freetown 00232, Sierra Leone; siamoreniketengbe@outlook.com (S.M.T.); hajamamakoh@yahoo.com (M.F.); samjokanu@yahoo.com (J.S.K.); lakoh2009@gmail.com (S.L.); spllenz54321@gmail.com (S.S.T.K.K.); kamarakadijatunabie@gmail.com (K.N.K.); mtmannah@yahoo.co.uk (M.T.M.); 6Republic of Sierra Leone Armed Forces, HIV/AIDS/TB Control Program Coordinator, 34th Military Hospital, Wilberforce, Western Area Urban 00232, Sierra Leone; mmmoiwo@gmail.com; 7College of Medicine and Allied Health Sciences, University of Sierra Leone, Freetown 00232, Sierra Leone; fthomas@pharmacyboard.gov.sl; 8National Pharmacovigilance Center, Pharmacy Board of Sierra Leone, Freetown 047235, Sierra Leone; 9UNICEF, UNDP, World Bank, WHO Special Programme for Research and Training in Tropical Diseases (TDR), 1211 Geneva, Switzerland; zachariahr@who.int

**Keywords:** health care-associated infections, antimicrobial resistance, IPCAF, IPCAT, core components, WASH, quality of care, universal health coverage, operational research, SORT IT

## Abstract

Introduction: Infection prevention and control (IPC) is crucial to limit health care-associated infections and antimicrobial resistance. An operational research study conducted in Sierra Leone in 2021 reported sub-optimal IPC performance and provided actionable recommendations for improvement. Methods: This was a before-and-after study involving the national IPC unit and all twelve district-level secondary public hospitals. IPC performance in 2021 (before) and in 2023 (after) was assessed using standardized World Health Organization checklists. IPC performance was graded as: inadequate (0–25%), basic (25.1–50%), intermediate (50.1–75%), and advanced (75.1–100%). Results: The overall IPC performance in the national IPC unit moved from intermediate (58%) to advanced (78%), with improvements in all six core components. Four out of six components achieved advanced levels when compared to the 2021 levels. The median score for hospitals moved from basic (50%) to intermediate (59%), with improvements in six of eight components. Three of four gaps identified in 2021 at the national IPC unit and four of seven at hospitals had been addressed by 2023. Conclusions: The study highlights the role of operational research in informing actions that improved IPC performance. There is a need to embed operational research as part of the routine monitoring of IPC programs.

## 1. Introduction

Infection prevention and control (IPC) measures at health facilities are a central pillar for protecting patients, visitors, and health workers from acquiring healthcare-associated infections (HAIs) [1]. The World Health Organization (WHO)’s Global Report on IPC (2022) revealed that good IPC programs could reduce 70% of HAIs [2]. Also, IPC is a strategic pillar of the Global Action Plan to tackle antimicrobial resistance (AMR), the logic being that “one prevented infection is one antibiotic treatment avoided” [3].

An effective IPC program is essential to ensure the quality of universal health coverage and health system resilience, especially in low- and middle-income countries with high rates of HAIs [4,5,6,7]. However, a global assessment of the IPC program (2020-21) showed that about 11% of countries did not have a national IPC program, and in 54% of countries, the IPC program was only implemented in a few health facilities [2,8]. This prompted a resolution during the 75th World Health Assembly in May 2022 requesting that WHO develop a global strategy for IPC, in consultation with member states and regional economic integration organizations [9].

The WHO guideline on “Core Components of Infection Prevention and Control at the National and Acute Health Care Facility Levels” recommends the optimal establishment of six IPC core components at the national level and eight at the health facility level [10]. In addition, the WHO developed standardized checklists with a percentage scoring system (0% to 100%) to assess the IPC performance and facilitate improvements [11,12]. A recent (2022) global survey using the WHO checklists reported that IPC performance scores were low in low-income countries and public health facilities [13]. Even individual studies from low-income countries like Tanzania, Liberia, Uganda, Pakistan, and Bangladesh have reported low IPC performance scores [14,15,16,17,18].

In countries like Sierra Leone, with a high risk of infectious diseases, establishing a good IPC program is paramount to prevent the transmission of infections of global concern such as pandemic influenza, Ebola virus disease (EVD) and other viral hemorrhagic fevers, and of recent concern, COVID-19. During the 2014–2015 EVD outbreak in West Africa, Sierra Leone alone reported about 14,000 cases and 3955 EVD deaths, including those of 221 health workers [19,20]. The rapid transmission of EVD among healthcare workers during the 2014–2015 EVD outbreak could be attributed to inadequate IPC practices. In 2015, the national IPC unit was established within the Ministry of Health and Sanitation (MoHS), and the IPC program was implemented in health facilities [20]. The strategic plan to combat AMR in Sierra Leone emphasized the need to improve IPC measures as a means to stop the accelerating rate of antimicrobial resistance [21].

In 2021, Fofanah et al. conducted operational research study using the WHO standardized checklists to assess the IPC performance and gaps in the national IPC unit and district-level secondary public hospitals of Sierra Leone [22]. This study was designed and conducted as part of the Structured Operational Research Training Initiative (SORT IT) to tackle antimicrobial resistance led by TDR, the Special Program for Research and Training in Tropical Diseases [23,24]. The operational research was used to generate knowledge on interventions, strategies, or tools that can enhance the quality, effectiveness, or coverage of IPC program studies [25]. Operational research is shown to be effective in helping policy makers to make evidence-based changes to policy or practice to improve program performance in different contexts [26,27,28,29].

The study by Fofanah et al. found that IPC performance was low, with an ‘intermediate’ level (58%) of performance in the national IPC unit and a ‘basic’ level (50%) in the district-level secondary public hospitals [22]. The priority gaps for low IPC performance at national and district-level hospitals were identified, and specific recommendations were made to address the identified gaps. In addition, through SORT IT, the capacity of the principal investigator (PI) was developed to prepare and use elevator pitches, plain language handouts, Lightning PowerPoints, and technical presentations for effective dissemination of study findings and recommendations to decision makers [23,30,31].

The TDR conducts post-SORT IT evaluation after 12 months of course completion as part of routine monitoring and evaluation. The PI of the previous study reported that the study findings were disseminated to decision makers. Some recommendations were translated into actions to improve IPC performance in national IPC units and hospitals (details provided in the methods section). Thus, we aimed to assess whether there was any improvement in the IPC performance scores across core components in April 2023 compared to those reported by Fofanah et al. [22] in June 2021 for the national IPC unit and the 12 district-level secondary public hospitals. We also assessed whether the gaps identified in 2021 were bridged by 2023.

## 2. Materials and Methods

### 2.1. Study Design

This was a before (June 2021) and after (April 2023) comparison of the data from the cross-sectional IPC assessments conducted routinely by the IPC program of Sierra Leone.

### 2.2. Study Setting

#### 2.2.1. General Setting

Sierra Leone is a West African country on the Atlantic coast bordering Guinea and Liberia. The Figure 1 below shows the geographic map of Sierra Leone which is made up of sixteen districts and five regions, and its population is around 8 million [32]. Tertiary hospitals (6), district-level secondary hospitals (4 regional hospitals and 8 district hospitals), other secondary hospitals (11), and peripheral health units (PHUs) provide services under the public healthcare system of the country [33]. From 1991 to 2002, Sierra Leone struggled through a civil conflict that disrupted the healthcare infrastructure. The resource-limited health system was also challenged by several infectious disease outbreaks of pandemic influenza, COVID-19, EVD, Lassa fever, and other high-risk infectious diseases [20]. 

Like Sierra Leone, the neighboring countries of Guinea and Liberia were also equally affected by multiple infectious disease outbreaks and the West African Ebola outbreak in 2013 to 2016. It is also true that these three countries share similar health system organizations and performances, including a shortage of healthcare workforce, information and surveillance research, medical products and technologies, health financing, and leadership. Following the EVD outbreak, WHO developed Infection Prevention and Control Recovery Plans for the three countries. It was reported that all the three countries made progress in terms of IPC and Water, Sanitation, and Hygiene (WASH) during the emergency response phase. Thus, the international agencies called for consolidating those gains in IPC by shifting from a vertical IPC Ebola response approach to an integrated and sustainable IPC program. This led to the establishment of IPC national and sub-national structures in these countries.

#### 2.2.2. The IPC Program in Sierra Leone

The outbreak of EVD in 2014–2015 exposed the deficiencies in IPC at health facilities, leading to the establishment of a national IPC unit by MoHS, with technical and funding support from WHO and other public health partners [20]. This unit is mandated to provide leadership, coordinate activities, provide training, and supervise the optimal implementation of IPC programs in health facilities. The unit consists of a national IPC coordinator and seven supporting IPC officers, with assigned tasks for supporting IPC implementation in health facilities.

At the hospital level, IPC programs are established, and designated IPC personnel/focal points coordinate IPC activities. The IPC committee in each health facility is responsible for implementing the national IPC policy and guidelines in their facilities. The national IPC program provides responsible personnel, accountable for action points, who also support health facilities in the optimal implementation of IPC.

#### 2.2.3. The IPC Performance Assessments

The standardized WHO checklists are used for routine IPC performance assessments of the national IPC unit and hospitals. IPC performance is assessed by the personnel from the national IPC unit and WHO, in consultation with the IPC focal points using checklists in paper form.

The National Infection Prevention and Control Assessment Tool (IPCAT) is used for the performance assessment of the national IPC unit (Appendix A) [11]. The IPCAT has 112 items, with ‘yes’ or ‘no’ responses for each item. The items are divided into six core components: (i) IPC program, (ii) IPC guidelines, (iii) IPC education and training, (iv) HAI surveillance, (v) IPC multimodal strategies for implementation, and (vi) IPC monitoring/audit of IPC practices. The total number of ‘yes’ responses from all the listed items under the core component is used to calculate a percentage score for each component. The scores can also be deduced for the subcomponents within each of the core components. The overall percentage score is derived using the total number of items with a ‘yes’ response out of the 112 items in the checklist [11].

The Infection Prevention and Control Assessment Framework (IPCAF) is used to assess the performance at the health facility level (Appendix A) [12]. The IPCAF checklist consists of 81 items, split across eight core components. The first six core components of the IPCAF are the same as those of the IPCAT; the additional two are ‘Workload, staffing and bed occupancy’ and ‘Built environment, materials and equipment for IPC at facility level’. In addition, the IPCAF percentage scoring system is the same as that of IPCAT.

#### 2.2.4. Dissemination of Findings of the Operational Research Study

As mentioned earlier, Fofanah et al. [22] conducted a study to assess the IPC performance of the national IPC unit and district-level secondary public hospitals. The PI and the co-investigators disseminated the study findings and the recommendations to the decision makers and key stakeholders involved in implementing the IPC program in the country. The published article and the dissemination materials developed during module 4 (module on communicating research findings) of the SORT IT course were used for dissemination. The dissemination details reported by the PI are provided in Table 1.

#### 2.2.5. Recommendations Made and Actions Taken

Following the effective dissemination of information, some of the recommendations by Fofanah et al. [22] were translated into actions. In Table 2, we have mapped the recommendations made and actions taken, as reported by the PI of the previous study.

### 2.3. Study Inclusion and Period

We included data from the routine IPC performance assessment of the national IPC unit and all the twelve district-level secondary public hospitals of Sierra Leone, conducted in April 2023. In addition, to compare the IPC performance with that in June 2021, we included the data from the IPC assessments of the same facilities reported in the previous study [22].

### 2.4. Data Variables and Sources

In April 2023, the IPC team conducted an IPC performance assessment using the paper-based WHO checklists, as was accomplished in 2021. In 2022, the national IPC unit made minor modifications to the WHO IPCAF checklist to adapt it to the country context and started using an adapted checklist for assessment. However, during the IPC performance assessment conducted in April 2023, the IPC team used the WHO IPCAF.

The IPC team entered the data from the paper-based checklist into a customized Microsoft Excel (Microsoft Corporation, Redmond, WA, USA, 2018) database developed by the WHO for calculating scores for each component of the IPCAT and IPCAF checklists. The calculated scores from the assessment of the national IPC unit and twelve district-level secondary public hospitals were merged into one Excel database for analysis.

We included the IPCAT scores in each of the six core components and sub-components for assessing whether there was any improvement in the IPC performance at the national IPC unit between 2021 and 2023. Similarly, the name of the assessed facility and the scores for the eight core components and sub-components of the IPCAF tool were included for assessing improvement in the IPC performance at district-level secondary public hospitals.

### 2.5. Data Analysis

Microsoft Excel (Microsoft Corporation, 2018) was used for data analysis. Each of the six core components of the IPCAT and eight core components of the IPCAF boast a maximum score of 100. The overall IPC performance score for the assessed unit or hospital was obtained by adding the scores in each of the core components. The maximum IPC performance score for the national IPC unit using IPCAT was 600 (six core components), and for hospitals using IPCAF, it was 800 (eight core components). The median scores were calculated to summarize the IPC performance in the twelve district-level secondary public hospitals.

A percentage IPC performance score was calculated (score obtained divided by maximum score multiplied by 100) for the overall score and scores for each core component and subcomponent. The percentage scores ranged from 0% to 100%. The IPC performance was graded based on the obtained percentage score: (i) inadequate (0–25%), (ii) basic (25.1–50%), (iii) intermediate (50.1–75%), and (iv) advanced (75.1–100%) level.

The change in the percentage IPC scores was calculated for each core component by subtracting the percentage score from 2021 from that of 2023. The radar charts were used to depict the percentage scores in the core components at the national IPC unit and district-level secondary hospitals during 2021 and 2023. The IPCAT and IPCAF sub-components with inadequate scores (≤25%) were considered as gaps, and these were compared between the scores from the 2021 and 2023 assessments.

## 3. Results

### 3.1. Assessment of IPC Performance at the National IPC Unit

#### 3.1.1. Change in IPC Performance Score between 2021 and 2023

The overall IPC performance improved from the intermediate (58%) to the advanced level (78%), with a 20% absolute improvement in the percentage score (Table 3).

There was an increase in the IPC percentage scores in each of the six core components. Four out of six core components achieved the advanced level in 2023, in contrast to only “IPC Guidelines” which had already achieved this status in 2021. Three additional core components which achieved the advanced level in 2023 were “IPC program” (88%), “Multimodal strategies” (83%), and “Monitoring/audits and feedback” (83%). In spite of more than 20% absolute improvement in “HAI surveillance” (27% to 53%), the component only reached the intermediate level (Figure 2).

#### 3.1.2. Gaps in IPCAT Sub-Components in 2021 and 2023

Table 4 shows the percentage scores for the 20 IPC sub-components in 2021 and 2023. In 2023, 13 of 20 sub-components had reached the advanced level, in contrast to only six in 2021. None of the sub-component’s scores showed a decline between the two assessments. Of the four sub-components which had inadequate scores (≤25%) and were considered as gaps in 2021, only one (“Monitoring of training and education”) remained as a gap in 2023.

### 3.2. Assessment of IPC Performance at District-Level Secondary Public Hospitals

#### 3.2.1. Change in IPC Performance Score between 2021 and 2023

The median overall IPC performance improved from basic (50%) to intermediate level (59%), with a 9% absolute improvement in percentage score (Table 5). Out of 12 hospitals, only one remained at the basic level (all others were at the intermediate level) in 2023, compared to seven during 2021 (Appendix A).

In district-level secondary public hospitals, there was an increase in the median IPC percentage scores in six of the eight core components between 2021 and 2023. Only the “IPC Program” component moved from the basic to the intermediate level, and the “IPC Guidelines” moved to the advanced from the intermediate level.

Although the “Workload, staffing, and bed occupancy” remained at the basic level, there was a 15% increase in the score (30% to 45%). Similarly, the “Built environment, materials, and equipment for IPC” remained at the basic level, even with an increase of 7% in the percentage score (Figure 3). There was a slight decline in median percentage scores for “IPC education and training” (2%).

#### 3.2.2. Gaps in IPCAF Sub-Components in 2021 and 2023

Table 6 shows the percentage scores of the selected 25 IPC sub-components in 2021 and 2023. In 2023, 9 of the 25 sub-components reached the advanced level, in contrast to only 5 in 2021. Among the sub-components, only the “Frequency of IPC training” showed a decline in the score between 2021 and 2023. Out of the seven sub-components considered as gaps (scores ≤ 25%) in 2021, three remained as a gap, even in 2023. The sub-components that remained as gaps were: lack of senior facility leadership commitment and support with a budget allocated specifically for IPC activities; not being able to establish a multidisciplinary team for implementing IPC multimodal strategies at hospitals; and no need-based assessment for hospital staffing.

## 4. Discussion

This before-and-after study showed an improvement in IPC performance scores at the national IPC unit and district-level secondary public hospitals following operational research on IPC by Fofanah et al. [22]. There was a substantial improvement in IPC performance at the national IPC unit, which moved from the intermediate (58%) to the advanced (78%) level. On the other hand, although there was an improvement in the IPC performance score (50% to 59%) at the hospitals, none of the hospitals reached the desired advanced level. Except for the “IPC program” and “IPC guidelines”, there was little improvement in other IPC core components at the hospitals.

This study is important as it shows the role of operational research in informing decisions and actions for improving the IPC implementation in line with recommendations of the WHO global report regarding IPC [2]. Also, this study justifies the 75th World Health Assembly resolution that recommends operationalizing research to produce science-based evidence on IPC [9]. The study findings are primarily important to the Western Sub-Saharan African countries such as Sierra Leone, Guinea, and Liberia, which are vulnerable to outbreaks and have a high risk of death due to antimicrobial resistance [19,20,34].

This study has several strengths. First, as the national IPC unit and all the district-level public hospitals were simultaneously assessed, the findings are generalizable and likely to reflect operational realities of IPC implementation in Sierra Leone. Second, the use of the same standardized WHO checklists reduced the risk of social desirability bias. Third, implementation of IPC checklists by the same IPC officers in both assessments (2021 and 2023) improved the internal validity by limiting observer bias. Finally, we adhered to the STROBE (Strengthening the Reporting of Observational Studies in Epidemiology) guidelines for conducting and reporting the study [35].

The study has three limitations. First, we could not quantify the contribution of the operational research by Fofanah et al. on the decisions made and actions taken to improve the IPC performance. In-depth qualitative interviews with the decision makers would have provided a better insight into the utilization of operational research in decision making and the root causes for the improvements in IPC performance. This merits further research. Second, as all the district-level hospitals were included in the study by Fofanah et al., we did not have a concurrent counterfactual (comparison group) to establish the independent effect of operational research on improving IPC performance. Third, as the IPC performance was reassessed in a relatively short time after the dissemination of the findings of Fofanah et al., some actions are still ongoing, and the full effect is yet to be manifested.

The study has six important implications for stakeholders involved in implementing an IPC program in low- and middle-income countries. First, the IPC performance scores improved at the national IPC unit and district-level secondary public hospitals following operational research. We believe that operational research contributed to this improvement in IPC performance, as six decision makers of IPC implementation (co-investigators of this study) acknowledged that the findings of operational research by Fofanah et al. galvanized them to make decisions and take actions.

The enabling factors for such research uptake included: (1) the relevance of the research topic; (2) the early engagement of six decision makers of IPC implementation in the conduct and reporting of research findings, thereby creating ownership of these research findings and the responsibility to act on the recommendations; (3) training of the PI of the previous study in research communication [30] and effective dissemination of study findings to the national IPC committee and other key stakeholders; and (4) the PI of the previous study being the WHO IPC focal point; as a result, he worked closely with the national IPC committee and was able to encourage research uptake and provide technical support for implementing recommendations. This experience highlights the importance of “local research, with local ownership, for local solutions”, as well as the importance the early engagement of decision maker in conceptualizing and conducting the study [36,37].

Second, there was a remarkable improvement in IPC performance at the national IPC unit. Sierra Leone now has a national IPC unit functioning at an advanced level, in contrast to most other low-income countries with non-functional national IPC programs [2,8,38]. This progress in the national IPC unit has a positive trickle-down effect on the IPC implementation in health facilities, as effective leadership, policies, action plans, and a framework for implementation exist. However, despite active advocacy in line with a recommendation from Fofanah et al., the national IPC program has yet to receive a dedicated budget. Globally, about three-fourths of the countries with national IPC programs do not have a dedicated budget [38]. A five-year funded national IPC action plan currently being developed with support from the WHO and other partners is a step forward for the creation of a dedicated budget. However, there is a need for continued efforts to present a funded national IPC action plan to the MoHS and acquire a dedicated budget for IPC programs across all levels.

Third, acknowledging the basic performance in the “Multimodal Strategy” and “HAI Surveillance”, the National IPC unit has concentrated on improving the performance of these core components. To improve the “Multimodal Strategy”, the national IPC unit facilitated the sub-national implementation of IPC improvement interventions and made effective linkages with other units of MoHS, including AMR, WASH, and Quality of Care. This is laudable, as studies have shown that an effective “Multimodal Strategy” can increase compliance with IPC interventions and reduce HAIs [39,40,41]. The national HAI surveillance strategy is being formulated with clear objectives and an implementation plan. Along with a national HAI surveillance strategy, the prospects of renovating laboratories under the AMR Fleming Fund grant to Sierra Leone is reassuring [42]. As a priority, the national IPC unit should finalize the HAI surveillance strategy and start implementing HAI surveillance.

Fourth, even though there was progress in IPC performance at district-level secondary public hospitals, none of the hospitals reached the desired advanced level. However, the progress from the basic to the intermediate level is satisfactory, as the WHO Global survey showed that the average IPC performance in the health facilities of low-income countries is at the basic level [13]. This progress is encouraging, as public hospitals of Sierra Leone still face infancy challenges, such as poor infrastructure and staffing, lack of IPC expertise, and weak financial resource allocation [13]. The improvement in IPC performance at district-level hospitals was mainly due to the implementation of recommendations such as: (1) ‘terms of reference’ for full-time IPC focal points; (2) distribution of updated national IPC guidelines and standard operating procedures across all hospitals; and (3) adopting and implementing IPC guidelines, including a well-defined monitoring plan with clear goals, targets, and activities.

Fifth, though recommended by Fofanah et al., no efforts were made to increase the number of hospital healthcare workers in line with the needs assessment. The country has no standards or guidelines for staff needs assessment for district-level secondary hospitals. Thus, the national IPC unit needs to follow up with MoHS to formulate and conduct staff needs assessments to improve the hospital health workforce. Similarly, the progress in “Built environment, materials, and equipment for IPC at the facility level” is suboptimal. However, WASH-FIT in-depth assessment to investigate and quantify the needs for effective WASH implementation in hospitals has been conducted. As WASH-FIT needs assessment helps hospitals to identify specific requirements to improve WASH practices [43,44], IPC focal points and hospital managers should leverage information from this assessment and actively advocate for fund allocation to fix these deficiencies.

Finally, given the utility of operational research in improving IPC performance through actionable recommendations, there is a need for embedding operational research in the routine monitoring of the IPC program. In line with the agenda of the national IPC action plan to prioritize operational research capacity-building activities, the IPC focal points in the hospitals can be trained through operational research programs like SORT IT. This would enable the IPC focal points to systematically identify and tackle local challenges in implementing IPC at hospitals and contribute to the global agenda of the 75th World Health Assembly to improve IPC [9].

## 5. Conclusions

This before-and-after study using standardized WHO checklists showed that the IPC performance improved in 2023 following the implementation of recommendations stemming from an operational research study in 2021. The active dissemination of relevant operational research findings and subsequent actions moved the IPC performance substantially from the intermediate to the advanced level at the national IPC unit and from the basic to the intermediate level at the district-level hospitals. This study highlights the importance of embedding operational research as part of routine IPC monitoring and its contribution to informed decision making. Further research is needed to explore the efficiency and the cost-effectiveness of improving the performance of the IPC program in the control and prevention of HAI and AMR.

## Figures and Tables

**Figure 1 tropicalmed-08-00376-f001:**
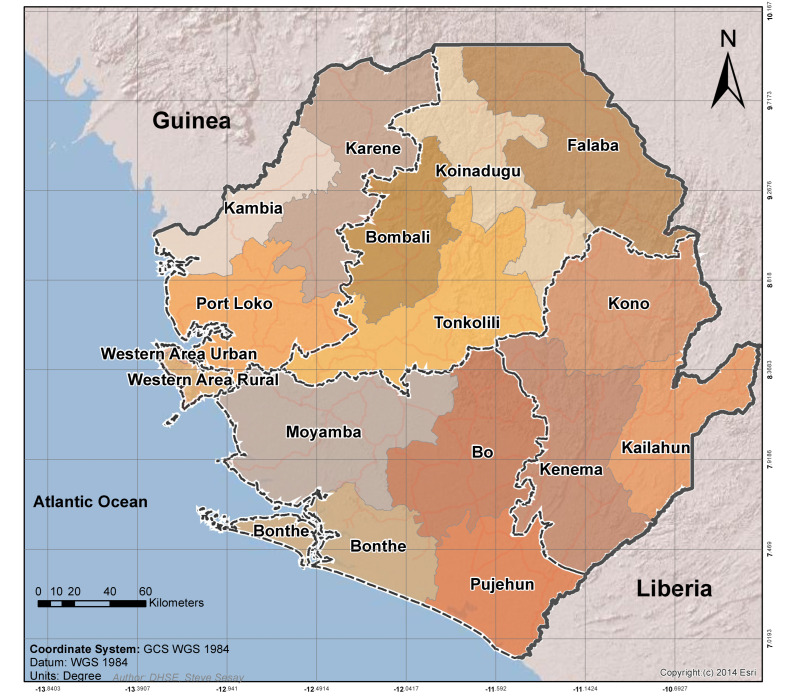
Map of Sierra Leone showing the 16 districts and the boundaries with Guinea and Liberia.

**Figure 2 tropicalmed-08-00376-f002:**
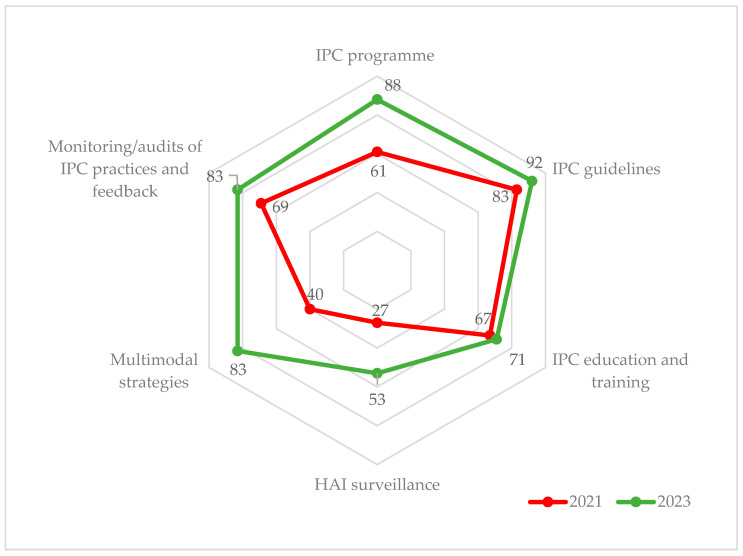
IPC percentage scores for National IPC unit assessment in June 2021 [22] and April 2023 in Sierra Leone. Note: the radar chart shows IPC components emanating from the center (0%) connected by lines expanding outwards (100%). The 2023 scores (green line) lie outside the 2021 scores (red line), indicating an overall improvement in all the components of IPC. The maximum score for each component is 100, and percentages are calculated relative to the maximum score for the component. Abbreviation: IPC = infection prevention and control; HAI = healthcare associated infection.

**Figure 3 tropicalmed-08-00376-f003:**
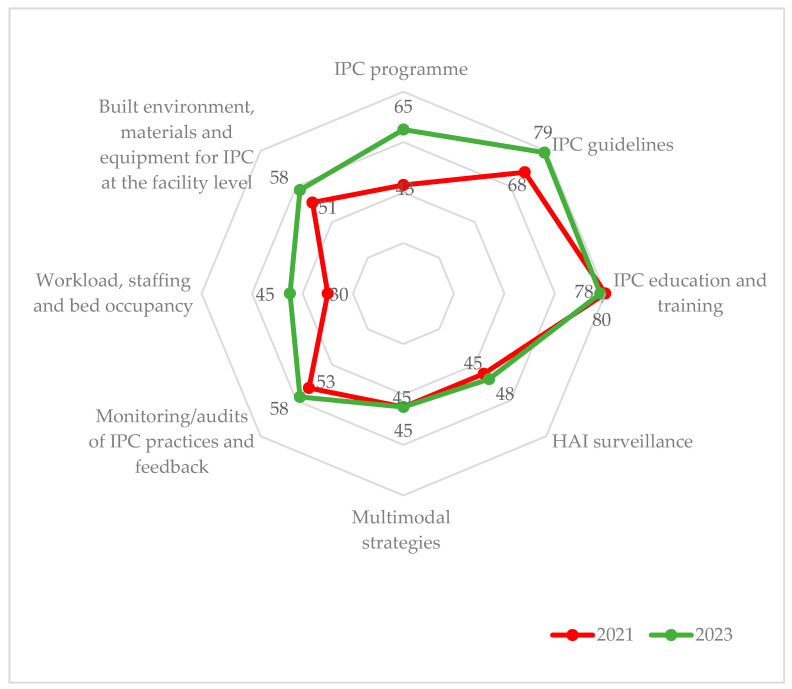
Median IPC scores in the twelve district-level secondary public hospitals during June 2021 [22] and April 2023 in Sierra Leone. Note: the radar chart shows IPC components emanating from the center (0%) connected by lines expanding outwards (100%). The 2023 scores (green line) lie outside the 2021 scores (red line) in six out of eight core components, indicating improvement in median IPC scores between the two assessments. Percentages are calculated relative to the maximum score for the component for each hospital. Later, the median percentage score for the twelve district hospitals was obtained. Abbreviation: IPC = infection prevention and control; HAI = healthcare associated infection.

**Table 1 tropicalmed-08-00376-t001:** Dissemination of findings and recommendations for improving IPC performance from the operational research study conducted by Fofanah et al. [22] in 2021 (Censored on 30 April 2023).

Mode of Delivery *	To Whom (Numbers ^$^)	Where	When
Three-min Lightning PowerPoint presentation	National IPC program (7)	National IPC unit	March 2022
	MoHS stakeholders (32)	National SORT IT module 4	April 2022
	MoHS stakeholders (16)	MoHS stakeholders meeting	July 2022
Published article [22]	Global and national IPC professional groups	Social media platforms—Whatsapp, Facebook, and LinkedIn	May 2022
	Hospital IPC focal points (16) and WHO AFRO IPC Team (2)	WhatsApp, email exchange, and during a consultative meeting to develop the national IPC action plan	February 2023
Ten-min technical PowerPoint presentation	Hospital IPC focal point (12)	IPC training	June 2022
	Hospital managers and IPC focal points (24)	National SORT IT dissemination meeting	November 2022
Plain language handouts [31]	Global and national IPC professional groups	Social media platforms—Whatsapp, Facebook, and LinkedIn	April 2022
	Researchers, AMR advocates, and community	WHO Sierra Leone website	March 2023

* Dissemination materials included a published article, a plain language handout, a three-minute Lightning PowerPoint presentation, a ten-minute technical presentation, and any other material; ^$^ The number of individuals attending the meeting. Abbreviations: SORT IT—Structured Operational Research Training IniTiative; IPC—infection prevention and control; MoHS—Ministry of Health and Sanitation; WHO AFRO: World Health Organization, Africa Region.

**Table 2 tropicalmed-08-00376-t002:** List of recommendations from the operational research study conducted in 2021 for improving IPC performance at national and facility levels and the status of these actions as of April 2023.

Recommendation	Action Status *	Details of Action (When)
Advocate for dedicated budget for IPC activities	Fully implemented	Activation of national IPC advisory committee headed by the deputy chief medical officer (public health), which advocated for a dedicated budget for an IPC program in October 2022.Funding secured for the national IPC unit to develop the national IPC action plan in March 2023.
Distribution of national IPC guidelines to health facilities	Fully implemented	The WHO printed 1200 copies of the updated national IPC guidelines, which were shared with all the health facilities in September 2022.
Dedicated time allocated to IPC staff at health facilities to adapt and implement IPC guidelines	Fully implemented	The WHO and National IPC unit shared ‘terms of reference’ for full-time IPC focal points in August 2022.
Clear goals, targets, and activities introduced in the monitoring framework for health facilities	Fully implemented	National IPC unit and WHO disseminated the updated monitoring frameworks in July 2022.IPC focal points of hospitals were trained on the monitoring frameworks in August 2022.
Increase the healthcare workforce	Not implemented	
Safe and sufficient water supplies	Not implemented	This recommendation was not directly implemented. However, in December 2022, the WASH manager (co-investigator in the previous study), with support from the WHO and national WASH and IPC unit, implemented the WASH-FIT in-depth assessment to investigate and quantify the needs for effective WASH implementation in hospitals. Based on WASH-FIT assessment, the specific actions at facility level were recommended.
Adequate numbers of functional toilet facilities		
Facilities for sterilization and disinfection		
Waste disposal		
Supply of consumables such as soap, alcohol-based hand rub, and personal protective equipment	Partially implemented	MOHS has continued its efforts to increase the local production and uninterrupted supply of soaps and alcohol-based hand rub to health facilities.
Formulating HAI surveillance strategy	Partially implemented	The WHO acquired funds from US CDC to develop the first HAI surveillance strategy for Sierra Leone, which is ongoing.
Access to microbiological laboratories	Not implemented	

* Fully implemented—actions taken and no further work required in line with the recommendation; partially implemented—some actions taken, but there is need for further work in line with the recommendation; not implemented—no action taken. Abbreviations: IPC—infection prevention and control; WHO—World Health Organization; WASH-FIT—Water and Sanitation for Health Facility Improvement Tool; US CDC—United States Center for Disease Control and Prevention.

**Table 3 tropicalmed-08-00376-t003:** Change in the IPC performance score at the national IPC unit between June 2021 [22] and April 2023 in Sierra Leone.

IPC Components ^a^	2021		2023		% Change ^d^
	Grade ^b^	% Score ^c^	Grade ^b^	% Score ^c^	
IPC program	Intermediate	61	Advanced	88	27
IPC guidelines	Advanced	83	Advanced	92	9
IPC education and training	Intermediate	67	Intermediate	71	4
HAI surveillance	Basic	27	Intermediate	53	26
Multimodal strategies	Basic	40	Advanced	83	43
Monitoring/audits of IPC practices and feedback	Intermediate	69	Advanced	83	14
Overall score (%)	Intermediate	58	Advanced	78	20

Abbreviation: IPC = infection prevention and control; HAI = healthcare associated infection. ^a^ Maximum score for each component is 100, and for the cumulative, it is 600. ^b^ Grade: IPC performance in each component was graded based on the obtained percentage: (i) inadequate (0–25%), (ii) basic (25.1–50%), (iii) intermediate (50.1–75%), and (iv) advanced (75.1–100%) level. ^c^ Percentages are calculated relative to the maximum score for the component. ^d^ Percentage in 2023—percentage in 2021.

**Table 4 tropicalmed-08-00376-t004:** The percentage scores of the sub-components of IPCAT at the national IPC unit in Sierra Leone during June 2021 [22] and April 2023.

IPC Core Components	Sub-Components	2021 *	2023 *
i. IPC Program	Organization and leadership of the program	63%	75%
	Defined scope of responsibilities	71%	100%
	Linkages with other programs and professional organizations	50%	88%
ii. IPC Guideline	Development, dissemination, and implementation of national technical guidelines	100%	100%
	Education and training of relevant healthcare workers on IPC guidelines	67%	67%
	Monitoring of guideline adherence	100%	100%
iii. IPC Education and Training	Supporting and facilitating IPC education and training at the facility level	100%	100%
	National curricula and IPC training and education	100%	100%
	Monitoring of training and education	0%	0%
	Implementation of training and education	67%	83%
iv. HAI Surveillance	Coordination of surveillance at the national level	43%	43%
	National objectives of surveillance	20%	80%
	Prioritized HAIs for surveillance	17%	50%
	Methods of surveillance	67%	100%
v. Multimodal Strategies	National and sub-national coordination in support of local implementation of IPC improvement interventions	100%	100%
	National and sub-national facilitation in support of local implementation of IPC improvement interventions	60%	80%
	Program and accreditation linkages	0%	50%
vi. Monitoring/Audits of IPC Practices and Feedback	Monitoring/audit and feedback framework for IPC	50%	67%
	Monitoring/audit indicators	75%	100%
	Monitoring/audit and feedback process and reporting	83%	83%

Abbreviation: IPC = infection prevention and control; HAI = healthcare associated infection. * The % score for each component in the facility was calculated using IPCAF scores. The median of this score for the national IPC program is presented. 
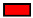
 Inadequate; 
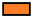
 Basic; 
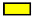
 Intermediate; 
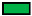
 Advanced.

**Table 5 tropicalmed-08-00376-t005:** Change in the median IPC percentage score in the twelve district-level secondary public hospitals between June 2021 [22] and April 2023 in Sierra Leone.

IPC Components ^a^	2021		2023		% Change ^d^
	Grade ^b^	Median % Score ^c^	Grade ^b^	Median % Score ^c^	
IPC program	Basic	43	Intermediate	65	22
IPC guidelines	Intermediate	68	Advanced	79	11
IPC education and training	Advanced	80	Advanced	78	-2
HAI surveillance	Basic	45	Basic	48	3
Multimodal strategies	Basic	45	Basic	45	0
Monitoring/audits of IPC practices and feedback	Intermediate	53	Intermediate	58	5
Workload, staffing, and bed occupancy	Basic	30	Basic	45	15
Built environment, materials, and equipment for IPC	Intermediate	51	Intermediate	58	7
Overall score (%)	Basic	50	Intermediate	59	9

Abbreviation: IPC = infection prevention and control; HAI = healthcare associated infection. ^a^ The maximum score for each component is 100, and for the overall score, it is 800. ^b^ Grade: IPC performance in each component was graded based on the obtained percentage: (i) inadequate (0–25%), (ii) basic (25.1–50%), (iii) intermediate (50.1–75%), and (iv) advanced (75.1–100%) level. ^c^ Percentages are calculated relative to the maximum score for the component for each hospital. Later, the median percentage score for the twelve district hospitals was obtained. ^d^ Percentage in 2023—percentage in 2021.

**Table 6 tropicalmed-08-00376-t006:** The median percentage scores of the sub-components of IPCAF in the twelve district-level secondary public hospitals in Sierra Leone during June 2021 [22] and April 2023.

IPC Core Components	Sub-Components	Median Percentage Score *	
		2021	2023
i. IPC Program	IPC program at facility	50%	75%
	Functional IPC committee	100%	100%
	Senior facility leadership commitment and support, with a budget allocated specifically for the IPC activities	0%	0%
ii. IPC Guideline	Expertise in IPC to develop or adapt guidelines	0%	100%
	Availability of IPC guidelines	57%	93%
	Consistent with national/international guidelines	100%	100%
iii. IPC Education and Training	Availability of personnel with the IPC expertise to lead IPC training	100%	100%
	Frequency of IPC training	67%	33%
	IPC training integrated in the clinical practice and training of other specialties	0%	50%
iv. HAI Surveillance	Surveillance as a defined component of IPC program	100%	100%
	HAI surveillance performed	14%	29%
	Methods of surveillance	40%	40%
v. Multimodal Strategies	Use of multimodal strategies to implement IPC interventions	100%	100%
	Multimodal strategy elements implemented in an integrated way	40%	40%
	A multidisciplinary team for implementing IPC multimodal strategies	0%	0%
vi. Monitoring/Audits of IPC Practices and Feedback	A well-defined monitoring plan, with clear goals, targets and activities	0%	100%
	Monitoring of IPC processes and indicators	44%	44%
	Feedback of auditing reports on the state of the IPC activities/performance	60%	80%
vii. Workload, Staffing, and Bed Occupancy	Assessment of hospital staffing needs	0%	0%
	Hospital bed occupancy	43%	54%
viii. Built Environment, Materials, and Equipment for IPC at the Facility Level	Water availability and access	42%	50%
	Functioning hand hygiene and sanitation facilities	67%	58%
	Patient placement and personal protective equipment (PPE) in health care settings	67%	67%
	Medical waste management, and sewage	50%	53%
	Decontamination and sterilization	50%	67%

Abbreviation: IPC = infection prevention and control; HAI = healthcare associated infection. * Percentages are calculated relative to the maximum score for the sub-component for each hospital. Later, the median percentage score for the twelve district hospitals was obtained. 
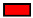
 Inadequate; 
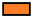
 Basic; 
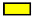
 Intermediate; 
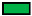
 Adequate.

## Data Availability

The metadata record of the data used in this paper is available at DOI 10.6084/m9.figshare.23574771 (accessed on 7–8 June 2023) under a CC BY 4.0 license. Requests to access these data should be sent to the corresponding author.

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
