# Peer review of "Improvement in Infection Prevention and Control Performance Following Operational Research in Sierra Leone: A Before (2021) and After (2023) Study"

_tropicalmed, 2023, doi:10.3390/tropicalmed8070376_

Round 1

Reviewer 1 Report

The authors conducted an operational research study in Sierra Leone in 2021 reported sub-optimal IPC performance and provided actionable recommendations for improvement. This was a before-and-after study involving the national IPC unit and all twelve district-level secondary public hospitals. IPC performance in 2021 (before) and in 2023 (after) was assessed using standardized World Health Organizations checklists. IPC performance was graded as: inadequate (0–25%), basic (25.1–50%), intermediate (50.1–75%), and advanced (75.1–100%). The overall IPC performance in the national IPC unit moved from Intermediate (58%) to Advanced (78%), with improvements in all six core components. Four out of six components achieved advanced level compared to one in 2021. The median score for hospitals moved from Basic (50%) to Intermediate (59%), with improvements in six of eight components. Three of four gaps identified in 2021 at national IPC unit and four of seven at hospitals had been addressed by 2023. The study highlights the role of operational research in informing actions that improved IPC performance. There is need to embed operational research as part of routine monitoring of IPC programmes.

Add a map of Sierra Leone.

The interaction between this country and other surrounding countries should be discussed.

How to extend your future work with integration with machine learning tools; see “Artificial intelligence for forecasting the prevalence of COVID-19 pandemic: an overview

Author Response

Dear Reviewer,

Regards.

Reviewer 2 Report

MDPI Revision

Tropical Medicine and Infections Disease

"Improvement in Infection Prevention and Control performance following operational research in Sierra Leone: A Before (2021) 3 and After (2023) Study"

The article is relevant, but it needs to enrich the motivation and analysis of the study supported by operational research. I suggest acceptance of the article through major revisions.

1 – The abstract should provide better detail of the article, perhaps indicating the main results of the analysis performed, including limitations and future research.

2 - The main problem of the work concerns the literature review, limited to explain the process and data, but forgetting the literature basis, as Operational Research, that is present in the title. Considering the relevance of the topic and the impact of this journal, a literature review should be considered, trying to reflect the importance of the proposed analysis with strategical implementation in health environment. Also, I suggest the inclusion and explanation of the main related works, based in the use of operational research as decision-making aid in complex problems.

The literature review should be comprehensive and complete, citing relevant studies. In this sense, as an indication of reading, to create a better structuring of bibliometric analysis, I suggest the following studies:

https://www.sciencedirect.com/science/article/pii/S1877050922000527

https://www.mdpi.com/2227-9032/10/11/2147

3 -A better motivation for this survey should be provided, including expelling the main gaps your research will fill.

4 – In the discussion or conclusion sections, you should present more details about the study's limitations and future proposals for a more accurate analysis.

5 – The introduction section need to provide the paper structure.

6 - The conclusion is too short. Provide more detail about the study's limitations and future proposals for a more accurate analysis.

7 – Provide a general review of English grammar.

Moderate editing of English language required

Author Response

Dear Reviewer,

Regards.

Round 2

Reviewer 1 Report

Accept.

OK

Reviewer 2 Report

The authors provided all the improvements indicated in the review.

In this way, I indicate the approval of the paper.

The english style is ok.